# Emerging Role of Circulating Tumor Cells in Gastric Cancer

**DOI:** 10.3390/cancers12030695

**Published:** 2020-03-15

**Authors:** Phung Thanh Huong, Sanjeev Gurshaney, Nguyen Thanh Binh, Anh Gia Pham, Huy Hoang Nguyen, Xuan Thanh Nguyen, Hai Pham-The, Phuong-Thao Tran, Khanh Truong Vu, Nhuong Xuan Duong, Claudio Pelucchi, Carlo La Vecchia, Paolo Boffetta, Hung D. Nguyen, Hung N. Luu

**Affiliations:** 1Department of Biochemistry, Hanoi University of Pharmacy, Hanoi 10000, Vietnam; huongpt@hup.edu.vn; 2Cancer Division, Burnett School of Biomedical Science, College of Medicine, University of Central Florida, Orlando, FL 32827, USA; gurshaneys@gmail.com; 3Department of Pharmaceutical Management and Economics, Hanoi University of Pharmacy, Hanoi 10000, Vietnam; binhnt@hup.edu.vn; 4Department of Surgical Oncology, Viet-Duc University Hospital, Hanoi 10000, Vietnam; phamgiaanh@gmail.com (A.G.P.); hoangnt35@gmail.com (H.H.N.); drxuan.vd@gmail.com (X.T.N.); 5Department of Pharmaceutical Chemistry, Hanoi University of Pharmacy, Hanoi 10000, Vietnam; hpham.phd@gmail.com (H.P.-T.); thaotp119@gmail.com (P.-T.T.); 6Department of Gastroenterology, Bach Mai Hospital, Hanoi 10000, Vietnam; vtruongkhanh@gmail.com; 7Department of Gastroenterology, 103 Hospital, Hanoi 10000, Vietnam; nhuongdx171@yahoo.com.vn; 8Department of Clinical, Sciences and Community Health, University of Milan, 20133 Milan, Italy; claudio.pelucchi@unimi.it (C.P.); carlo.lavecchia@unimi.it (C.L.V.); 9Icahn School of Medicine at Mount Sinai, Tisch Cancer Institute, Division of Hematology and Medical Oncology, New York, NY 10029, USA; paolo.boffetta@mssm.edu; 10Department of Epidemiology, University of Pittsburg Graduate School of Public Health, Pittsburg, PA 15261, USA; 11Division of Cancer Control and Population Sciences, UPMC Hillman Cancer Center, University of Pittsburgh Medical Center, Pittsburgh, PA 15232, USA

**Keywords:** circulating tumor cells, gastric cancer, diagnostic, prognostic, treatment

## Abstract

With over 1 million incidence cases and more than 780,000 deaths in 2018, gastric cancer (GC) was ranked as the 5th most common cancer and the 3rd leading cause of cancer deaths worldwide. Though several biomarkers, including carcinoembryonic antigen (CEA), cancer antigen 19-9 (CA19-9), and cancer antigen 72-4 (CA72-4), have been identified, their diagnostic accuracies were modest. Circulating tumor cells (CTCs), cells derived from tumors and present in body fluids, have recently emerged as promising biomarkers, diagnostically and prognostically, of cancers, including GC. In this review, we present the landscape of CTCs from migration, to the presence in circulation, biologic properties, and morphologic heterogeneities. We evaluated clinical implications of CTCs in GC patients, including diagnosis, prognosis, and therapeutic management, as well as their application in immunotherapy. On the one hand, major challenges in using CTCs in GC were analyzed, from the differences of cut-off values of CTC positivity, to techniques used for sampling, storage conditions, and CTC molecular markers, as well as the unavailability of relevant enrichment and detection techniques. On the other hand, we discussed future perspectives of using CTCs in GC management and research, including the use of circulating tumor microembolies; of CTC checkpoint blockade in immunotherapy; and of organoid models. Despite the fact that there are remaining challenges in techniques, CTCs have potential as novel biomarkers and/or a non-invasive method for diagnostics, prognostics, and treatment monitoring of GC, particularly in the era of precision medicine.

## 1. Introduction

According to data from the International Agency for Research on Cancer, over 1 million new cases of gastric cancer (GC) were found in the past year, and GC was determined to be the third highest cause of cancer-related mortalities worldwide [1,2]. The overall five-year survival rate of GC after curative surgery is approximately 30%. There is a great variation of GC survival rate from developed to developing countries [3], as a patient’s prognosis is highly dependent on the stage of the tumor found during surgery. In areas where endoscopic screenings are regularly conducted, such as Japan, a significantly high five-year survival rate is achieved due to the increased possibility for early tumor resection [4]. This survival rate can also drop to less than 5% if the patient is diagnosed at later stages (IIIB to IV) (https://www.cancer.net/). Reliable screening and early diagnosis, therefore, play a critical role in the management of GC and patient survival. Despite recent advancement in GC diagnosis and treatment, prognostic outcomes for patients remain quite poor [5]. The asymptomatic characteristics at early stages, high recurrent risk, and rapid metastasis after operation are the main problems that hinder effective diagnosis and treatment of GC. 

To date, several biomarkers have been successfully identified and widely used for GC diagnosis and prognosis, including carcinoembryonic antigen (CEA), cancer antigen 19-9 (CA19-9), and cancer antigen 72-4 (CA72-4) [5]. However, those biomarkers display less than 40% positivity rates, and their sensitivities and specificities are modest [5,6]. In this regard, circulating tumor cells (CTCs) emerge as promising markers to monitor treatment responses and to provide progression information in a real-time manner [7,8,9,10]. 

CTCs are cells that dissociate from the primary tumor and travel through the vasculature. Prior studies have shown that in many cases, the spread and migration of CTCs are heavily responsible for the establishment of distant metastases [11]. CTCs are present in body fluids well before metastasis and they could potentially be used as a predictive indicator of future metastatic formations [11]. Therefore, the use of CTCs for cancer screenings is really promising, as the routine profiling and detection of CTCs from the blood could lead towards the development of a far more efficient, noninvasive system for detection of tumors. Additionally, CTCs exhibit certain properties and unique molecular traits of the primary tumor that could provide valuable insights into possible mechanisms of cancer treatment [12,13,14].

Currently, CTCs have been approved by the FDA as a prognostic biomarker for monitoring patients with breast, prostate, and colorectal cancer [15]. However, the clinical utility of CTCs in GC is still limited. Given the importance of the topic, we performed this review with an aim to provide an up-to-date, comprehensive picture of CTCs in GC, from a biological standpoint to their clinical applications. We discussed the challenges and future perspectives of CTC use in the diagnosis, prognosis, and treatment of GC.

## 2. Landscape of CTCs

### 2.1. CTC Migration

The process of CTC migration is critical towards understanding their behavior in circulation. In the vasculature, CTCs are exposed to attacks from the immune system, high shear stress, and oxidative stress, which pose severely harsh conditions for CTC survival [16,17]. The vast majority of CTCs fail to survive in this foreign environment. However, few CTCs are able to survive and contribute towards the formation of distant metastases, raising the questions of how they are able to do this and what molecular mechanisms are involved.

The traditional model of CTC migration relies heavily upon the epithelial to mesenchymal transition (EMT) process. A major defining characteristic of CTCs in the epithelial state is the expression of e-cadherin, a surface protein heavily involved in the formation of tight junctions and responsible for cellular adhesion [18,19,20]. Consequently, CTCs are inter-locked in a network of intracellular adhesions and lack of migration capabilities. In their mesenchymal state, however, the cells exhibit far greater invasive and migratory properties and are less prone to apoptosis. The increased expression of transforming growth factor β (TGF-β) and several transcriptional factors play critical roles in the induction of EMT and lead to the downregulation of e-cadherin, along with other epithelial markers like alpha and beta catenin [21,22,23,24,25]. The overexpression of extracellular matrix (ECM) proteins such as vimentin and fibronectin results in the induction of certain mesenchymal properties [26,27]. Additional EMT-induced modifications that are likely to help the CTCs adapt in the harsh microenvironments, such as the upregulation of cluster of differentiation 47 (CD47) for increased defense against immune attacks or the downregulation of chaperone calreticulin to dodge immune-surveillance [28]. Further, TGF-β has been shown to induce increased coagulation, resulting in the formation of a platelet-rich microclot around CTCs which could physically protect them from the high shear stress in the vasculature as well as provide shielding from NK-mediated attack [29]. EMT has also been shown to activate survival pathways such as Akt and P13k signaling to provide resistance to apoptosis [29].

It is noteworthy that after EMT, CTCs are able to invade the blood vessels and lymph capillaries and migrate towards various organs [30,31]. Once the CTCs are in the vasculature, they can either establish metastases in distant organs or return to the primary tumor, a process known as self-seeding [16]. Self-seeding contributes towards metastatic development, because due to the leaky neovasculature of tumors, “self-seeding” CTCs in circulation would not require significant adaptations to infiltrate their host tumor [32]. Consequently, the host tumor would be enriched with aggressive CTCs that would lead to an increase in the overall metastatic potential of the tumor. It was found by Kim et al. in a study of MDA231 mammary tumors in mice that self-seeded tumors had a significant increase in the rate of tumor growth, along with increased angiogenesis and stromal recruitment [32]. 

At their target location, CTCs can accomplish infiltration by undergoing a mesenchymal-to-epithelial cell transition (MET), the reversion of EMT, allowing them to regain their cellular adhesion [33,34]. Yet, even after observing the various modifications that CTCs have undergone for migration, our understanding on how individual CTCs are able to survive in the vasculature and to establish metastases is yet to be elucidated. CTC clusters were generally ignored as there has been a speculation that their large size would enable them to rupture blood vessel walls [17,35]. However, CTC clusters have been observed in the peripheral blood, and it is known that CTC clusters are easier to resist to high shear stress, immune system attacks and anchorage-dependent apoptosis than individual cells [35,36]. Therefore, the migration of CTCs in clusters has gained attention as they could have a far greater impact on colonization and metastatic potential than individual CTCs.

The inner regions of CTC clusters include networks of densely connected cells connected by cell junctions that express various adhesion molecules, while the cells on the outer regions of CTC clusters exhibit far greater migratory properties and are receptive to extracellular inputs directing its migration [30]. Certain received inputs lead to the activation of selected pathways, including mitogen-activated protein kinase (MAPK), extracellular-signal-regulated kinase (ERK), etc., that result in leader cell-type selection in certain CTC subsets and inhibition of leader cell-type selection in the adjacent areas [30]. Unlike individual CTCs, CTC clusters do not completely differentiate from the epithelial to mesenchymal type. Instead, there exists a spectrum of a hybrid epithelial and mesenchymal phenotype. It is also suggested that EMT occurred in a morphogenetic manner, as certain cells in the outer region of CTC cluster tend to lose the majority of their epithelial profile and begin expressing mesenchymal markers, while follower cells are able to retain most of their epithelial phenotype [16,37,38]. 

It is therefore obvious that CTCs have evolved an extensive and nuanced process for migration. Many of the mechanisms behind the migration of CTCs depend on their highly plastic nature, which is observed in their adaptation to different phenotypes and characteristics according to their microenvironment. Yet, there are still many gaps in our understanding of CTCs migration and thus further research on this area is warranted.

### 2.2. CTCs in Circulation

Once CTCs enter the bloodstream, they have the potential to colonize to the distant organs and establish metastases or return to their primary tumor site(s). However, even with the extensive modifications they have undergone, most CTCs, whether they are circulating individually or in clusters, are unable to survive in the hostile environment of the vasculature before they can extravasate into surrounding tissues [13]. 

Different studies have demonstrated that the few survivable CTCs which have high metastatic potential, undergo various mechanisms to prolong survival while in circulation. For instance, Shliakhtunou et al. [39] found significant overexpression of the survivin gene in all 36 samples of primary breast carcinoma with malignant tumors compared with benign ones. Moderate and late-stage tumors, in particular, express survivin at a significantly high level. Survivin suppresses apoptosis by inhibiting caspase 6 and 8 activation and also provides a defense against NK cell cytotoxicity; therefore, its overexpression would provide a survival advantage towards CTCs [40]. Cytokines are also extensively involved with CTCs in the circulation. In a study by Huh et al., human melanoma cells were injected into mice intravenously [41]. Upon administration of human neutrophils, cancer cell retention in the lungs and metastatic growth significantly increased in comparison to non-neutrophil treated mice. Further, the secretion of cytokine IL-8 by melanoma cells attracts neutrophils and promotes B-integrin expression, which facilitates anchorage of the cells to the vascular endothelium [41]. Additionally, Divella et al. found that CTC clusters in patients with metastatic colorectal cancer were significantly associated with elevated levels of cytokines CXCL1 and TGF-beta in circulation [42]. Other studies have also demonstrated this correlation between elevated production of cytokines and CTCs [43,44,45]. Taken together, these findings suggest that CTCs exhibit a cytokine-mediated interaction which aids in CTC survival, transendothelial migration and metastatic development. 

It has also been observed that platelet interaction plays an important role in CTC survival. EMT-mediated platelet coating around the CTC surface provides a shield from natural killer (NK)-mediated tumor attack as well from the shear and dynamic stress in the bloodstream [46,47]. Furthermore, platelet depletion enhances the efficacy of NK cytolysis of CTCs [48]. Accordingly, Placke et al. [49] utilized immunoelectron microscopy, immunofluorescent staining, and other techniques to reveal that platelet coating of CTCs results in the transfer of platelet-derived major histocompatibility complex (MHC Class 1) to the cell surface of CTCs, allowing them to mimic the phenotype of a host cell and therefore escape from immunosurveillance. These results suggested that therapies targeting platelet production may have critical effects in reducing CTC survival and preventing metastasis formation. Additionally, conditions such as thrombocytosis associated with excessive platelet formation may be used as prognostic indicators for tumor malignancy and migration, as increased platelet accumulation stemming from these conditions may help facilitate CTC survival and hence promote metastatic formation [50].

Overall, in circulation, there are several mechanisms in place to protect CTCs from the harsh environment of the vasculature. Further understanding these mechanisms could provide valuable insight into possible therapies that can inhibit the progression of certain tumors.

### 2.3. Morphologic Heterogeneity of CTCs 

A critical property of CTCs are their heterogeneities. CTCs derived from the same tumor exhibit a wide variety of morphological traits, which include size, ratio of nucleus to cytoplasm, structural irregularities, and apoptotic phenotype. CTC heterogeneity is largely responsible for its metastatic capabilities, as greater diversity in CTC morphology allows clonal selection of CTCs at secondary tumor site [51]. Generally, conventional CTCs that are positive for cytokeratin and EpCAM and negative for CD45 have a larger size and nucleus-to-cytoplasm ratio than leukocytes [52]. They have a distinct cytokeratin signal in the cytoplasm around the nucleus that is larger than the surrounding white blood cell nuclei [13]. CTCs that have a more mesenchymal phenotype and a reduced EpCAM expression and cytokeratin signal exhibit a large and irregularly shaped nucleus [13]. Other CTCs may have a nucleus with no size difference than those of white blood cell (WBC), or they may be apoptotic, showing a disrupted nucleus and/or possibly cytokeratin stain while also expressing M30, a neoepitope heavily involved in carrying out apoptosis [13,53]. They may have significant implications in assessing metastatic potential of tumors in patients. Kallergi et al., using 56 CTC positive patients, demonstrated that apoptotic CTCs were found under significantly lower amounts in patients that had metastatic disease [54]. Additionally, they found that the percentage of apoptotic CTCs was lower in patients with advanced stages of disease compared to patients who had early stages. 

CTCs derived from the same tumor exhibit varying morphological and immunophenotypical characteristics. Given the few examples presented above of CTC heterogeneity, this diversity may have implications in the use of CTCs to analyze the effectiveness of certain therapeutic strategies for disease treatment [53].

## 3. Clinical Implication of CTCs in GC 

Although CTCs were discovered 150 years ago, the studies on clinical utilities of CTCs in GC has peaked recently, particular during the past five years. The systematic review in Table 1 shows that CTCs have significant values in the diagnosis, prognosis, and treatment management of GC at various stages. 

### 3.1. Diagnostic Potential of CTC in GC 

Similar to other types of cancer, early detection is one of the most important challenges to GC since 80% of patients are asymptomatic during the early stages [69]. Recently, the role of CTCs in GC diagnosis has markedly gained attention. A meta-analysis by Tang et al. [70] recommended that CTC detection alone could not be used for GC screening because of poor sensitivity. However, with a high positive likelihood ratio and especially the specificity of 0.99 and the area under the curve (AUC) of 0.97 in the receiver operating characteristic (ROC) curve, CTCs detection could be useful for the confirmation of GC. Intriguingly, also according to Tang et al. [70], from 1997 until 2012, there was an annually increased trend of sensitivity, suggesting that the advancement in detection technology might help improving the diagnostic accuracy of CTCs. Indeed, using a centrifugal microfluidic system with a new fluid-assisted separation technique, Kang et al. [61] showed that the sensitivity and specificity of CTCs in GC detection were 85.3% and 90.3%, respectively. A cut off ≥ 2 CTCs per 7.5 mL of blood was useful for differentiating GC patients from healthy controls.

In addition to CTC enumeration, CTC molecular profiles can be used for GC diagnosis as well. A combination of cytokeratin 19 and 20 from CTCs can detect GC with sensitivity and specificity of 87.5% and 94.7%, respectively, suggesting their use in the diagnosis of GC [71]. Recently, miRNAs are emerging as biomarkers in diagnosis and treatment of cancers in general and GC in particular [72,73]. A number of miRNAs from peripheral mononuclear cells could be used as surrogate markers to highlight GC patients. For instance, the ROC-AUC obtained with miR-421, a miRNA which is overexpressed in GC, targeting E-cadherin and caspase-3, thereby enhancing metastasis and attenuating apoptosis [74], is 0.77, with a positive detection rate of 72.5%, much higher than that of CEA, one of the most conventional tumor markers. A cut-off value of 7.075 is applied for miR-421 to detect GC [75]. Similarly, several oncogenic proliferation-triggering miRNA such as miR-543 [76] targeting Sirtuin 1 (SIRT1), miR-106a targeting tissue inhibitor of metalloproteinases 2 (TIMP2) [77], or miR-17 which targets a number of genes involving in tumorigenesis, disease progression, invasion, and metastasis of GC [78,79] have been suggested as candidate markers for GC diagnosis.

### 3.2. CTC Assay for Prognosis and Treatment Management of GC

In spite of advances in surgery in combination with chemo-radiotherapy in GC management, metastasis and relapse are still of great concern. Relevant assessment of malignant stages and prognosis are important to the success of such regimens. A meta-analysis on 2566 GC patients from Asia, Europe, and Africa showed a significant correlation between the detection incidence of CTCs and clinico-pathological parameters, including cancer stages, vessel and lymphatic invasion, and tumor histological differentiation. Moreover, there was a correlation between CTCs and disease-free survival (DFS) (hazard ratio—HR = 3.42, 95% confidence interval-CI: 2.39–4.91) as well as the overall survival (OS) (HR = 2.13, 95% CI: 1.13–4.03). Such results support the potential of using CTCs as a prognostic indicator in GC [80]. Similarly, in the study by Zheng et al. [56], positive CTC counts were more common for tumors with diffuse histologic type and distant metastasis, followed by significantly shorter progression-free survival (PFS) (HR = 2.03, *p* = 0.016). Accordingly, CTCs can be used as a biomarker for considering patients who need intensive treatment (Figure 1). 

Additionally, in a study by Li et al. [81], the post-therapy CTC level was used for monitoring therapeutic response in GC patients and predicting their prognosis. Following 6 weeks of chemotherapy, an unfavorable CTC count (≥ 3 CTCs per 7.5 mL) significantly correlated with the objective response rate (*p* = 0.02) and the disease control rate (*p* = 0.01) and furthermore could independently predict a shorter PFS and OS. In contrast, a continuously favorable CTC count or an early shift into a favorable CTC count following therapy could predict an improved prognosis. Similarly, in Liu’s report [59], only those patients with a low baseline CTC level or a decreased CTC level after the first cycle of chemotherapy remarkably benefited from the treatment. Those with a baseline CTC level of > 2 cells/5 mL blood, which was determined as an independent poor prognostic marker for PFS and OS, were spared from the adverse reactions of chemotherapy. The role of CTC count in the prognosis and prediction of chemotherapeutic response was also supported in metastatic GC patients [55]. In chemotherapy, not only can CTC count be used to predict recurrence risk after operative therapy, there is also a significant difference in the recurrence rates of those patients with different preoperative and postoperative CTC levels. A cut-off value of 1 CTC/7.5 mL was chosen for CTC measurement. Advanced GC patients with preoperative CTCs ≥1 and increased postoperatively had the poorest prognosis [82]. 

It should be noted that, not only the number of CTCs but their genotypes and phenotypes can affect the prognostic significance in GC (Figure 1). Szczepanik et al. [83] demonstrated that among the cytokeratin positive CTCs, only those with CD44+ phenotype were correlated with poor prognosis and metastasis. CD44 is a transmembrane glycoprotein that was one of the first molecules identified on GC stem cells and is now known to be a marker for cancer stem cells (CSCs) [84]. The superior malignancy of the CD44^(+)^/CD45^(−)^ gastric CTCs over the human GC cells was supported in a comparison by Yuan et al. [85]. Moreover, in an in vitro screening by Yuan et al. [85], such CTCs were found to be relatively sensitive to *5* fluorouracil, cisplatin, and paclitaxel, but relatively resistant to radiation, oxaliplatin, cetuximab, and trastuzumab. Along with the fact that CD44^( + )^/CD45^(−)^ CTCs show some degree of chemotherapy resistance, they have also been shown to exhibit other characteristics of cancer stem cells (CSC). For example, Grillet et al. generated ex vivo CTC cell lines and found that they were able to differentiate into distinct heterogenous lineages, indicating the presence of multipotent cell population resembling CSCs. Additionally, they also found significant expression of CD44 and aldehyde dehydrogenase, a CSC marker, in CTC cell lines. This indicates that CD44^(+)^/CD45^(−)^ CTCs functionally resemble or potentially are CSCs. Therefore, CTCs can mimic the genetic features of tumors at a given time and represent the current state of disease [86]. CTCs cultured in vitro can be used for drug sensitivity screening, a promising approach toward precision medicines. The idea of using CTC culture like an antibiogram has also been raised with other kinds of cancers, such as advanced metastatic colorectal cancer or breast cancer [87,88]. 

In precision medicine, patient stratification is essential to select those who respond to a treatment regimen. The response indicators are usually molecular biomarkers in biopsy samples from primary tumors. For instance, HER2 upregulation in primary tumors has been used as a biomarker for personalization of trastuzumab, a HER2 monoclonal antibody in advanced GC [89,90]. Intriguingly, Mishima et al. [58] recently demonstrated that HER2 amplification in CTCs was found even in advanced GC patients with HER2 negativity in tumor biopsy. Those patients received clinical benefits from trastuzumab which were comparable to those from the trastuzumab in combination with chemotherapy of the ToGA clinical trial [90]. Similarly, as reported by Abdallah et al. [67], three out of five nonmetastatic gastric adenocarcinoma patients with disease progression had HER2-negative primary tumors but HER2-positive CTCs. A significant CTC count drop in follow-up was seen for CTC-HER2-positive cases and CTC-plakoglobin-positive cases compared with CTC-HER2-negative cases and CTC-plakoglobin-negative cases. 

Genetic signatures of CTCs could reveal the metastatic capability and the malignancy of the cancer cells [14,91]. Mimori et al., using microarray analysis and RT-PCR on 810 GC patients, identified membrane type 1 matrix metalloproteinase (MT1-MMP) as a marker in CTCs for metastatic growth [92]. Further, Li et al. profiled CTCs in 31 patients with advanced GC [93] found that those with greater percentages of aneuploid CTCs had significantly poorer progression-free survival and overall survival. Similarly, Li et al. in karyotyping CTCs from patients with advanced GC found that those with chromosome 8 aneuploidy CTCs displayed decreased sensitivity and increased resistance to paclitaxel- and cisplatin-based therapy [94]. Such findings open a perspective of using CTCs as an alternative biopsy which is less invasive and more effective than the conventional tumor biopsy for GC prognosis and therapeutic management.

### 3.3. The Role of CTCs in GC Immunotherapy 

With several recent breakthroughs, immunotherapy is an important facet in the improvement of GC management. Until now, the two major approaches of immunotherapy that have been applied in GC clinical trials are adoptive cell immunotherapy and immune checkpoint inhibitor use [95]. The immune checkpoint blockade is progressing more rapidly with a promising response rate from pembrolizumab, a programmed cell death-1 (PD-1) inhibitor, in advanced GC patients [96]. Now, PD-L1 is the most common biomarker for PD-1/PD-L1 checkpoint blockade therapies. The FDA has approved four immunohistochemistry (IHC)-based assays for predicting therapeutic responses to PD-1/PD-L1 blockers. Nonetheless, the predictive value of PD-L1 IHC is still controversial, since in various types of cancer, including GC, some patients with PD-L1 negative tumors also respond to anti-PD-1 therapy [96,97]. Recently, Cheng et al. [65] found that in GC patients with negative PD-L1 in tumor tissue, the CTC Imaging flow cytometry signal was as varied as that of IHC staining, suggesting that expression of CTC PD-L1 is useful in the immunophenotypic differential diagnosis of tumors and thus can be a potential candidate for anti-PD-1/PD-L1 immune checkpoint therapy [98]. CTC level after checkpoint blockade therapy may be effective to predict poor prognosis in advanced GC [99]. When CTC PD-L1 has been recognized as a biomarker for therapeutic efficacy of immune checkpoint inhibition in non-small cell lung cancer (NSCLC), it is unclear whether it can be a predictive marker for GC immune checkpoint therapy [100,101]. Along this line, Yue et al. [62] demonstrated that PD-L1 level in CTCs can be a potential predictor for PD-1/PD-L1 blockade therapies in patients with advanced gastrointestinal tumors. The disease control (DC) rate in the patients with high CTC PD-L1 level was much higher than those with lower level. Additionally, several PD-L1 high CTCs at the baseline had predictive values for PFS. Moreover, the dynamic changes of PD-L1 positive CTCs can reflect real time response status. Indeed, in Yue’s study [62], the count changes of total CTCs, PD-L1 positive CTC, and PD-L1 high CTC significantly correlated with the clinical outcomes (Figure 1). This is an advantage of CTC PD-L1 over tumor PD-L1, which can only provide baseline information for treatment response prediction. Similarly, Cheng et al., using the CanPatrol CTC enrichment technique to validate the blood samples from 32 GC patients, also demonstrated that total CTC pool and CTC-PD-L1 are highly correlated with clinical outcome of checkpoint blockade therapy. In addition, high circulating PD-L1 expression in advanced GC patients is predictive of better five-year OS [102]. Altogether, these evidences support CTC PD-L1 expression as a prognostic factor for checkpoint blockade therapy efficacy. Moreover, recent advances in -OMICs technologies to support single-cell molecular profiling of CTCs and tumor cells using next-generation sequencing may also provide the pool of tools to validate predictive biomarkers including CTC PD-1 and PD-L1.

## 4. Challenge and Future Perspective 

### 4.1. Challenge

Despites promising perspectives of using CTCs as one type of liquid biopsy for GC diagnosis and prognosis, there are still remarkable challenges for translating them into clinical practice. Notably, there are controversies on the cut-off value of CTC positivity, since a wide range of CTC levels have been used in various studies [55,61,81,82]. Critical reasons for this include the unavailability of consensus in technical approaches ranging from sampling, storage conditions, and CTC molecular markers, along with a lack of relevant enrichment and detection techniques. To date, most of widely used CTC detection technologies, including the FDA-approved technology CellSearch, are based on epithelial markers (i.e., EpCAM, cytokeratins) and epithelial cell surface-associated glycoproteins (i.e., MUC-1) along with the absence of a leukocyte marker CD45. CTCs are, however, heterogeneous, especially in metastasis; as a result of EMT, more malignant CTCs with greater mesenchymal phenotypes have been found. Hence, it is difficult to detect the CTCs with mesenchymal phenotypes by the current technologies [103]. It is thus essential to develop better technologies which can detect a wide range of markers for various CTC subpopulations. 

### 4.2. Future Perspective

The current clinical utility of CTCs in GC is limited compared to broad perspectives of using such non-invasive biopsy for an insight of tumor, thereby leading to new approach of GC diagnosis, prognosis, and treatment. 

The major challenge in CTCs research and application is their precise detection and enrichment. There have been considerations on the advantages and disadvantages of single cell-analysis of CTCs over the tumor-derived cell-free DNA (cfDNA) in clinical application. The latter is more stable, easier to be isolated from blood and analyzed but requires an advance knowledge of the targets of interest, and its source cannot be clearly defined [104]. Therefore, a simultaneous analysis of both CTCs and cfDNA would enable better clinical significance. In fact, Keup found no difference of cfDNA isolated from whole blood and that from CTC-depleted blood, suggesting a solution of combined analysis [105]. On the other hand, the challenge of CTC single cell analysis could be attenuated by detecting CTC clusters or circulating tumor microembolies (CTM) instead of single CTCs. Using isolation by size of epithelial tumor cells (ISET), a technology based on differences in the size and shape of cancer cells and normal blood cells, Zheng et al. [60] efficiently detected CTCs/CTM in GC patients. CTM positivity, which was correlated with serum CA125 level, was an independent predictor of PFS and OS of stage IV patients. Given the stronger metastasis capability of CTMs over CTCs, CTMs can be developed into not only prognostic biomarkers, but also as therapeutic targets for anti-metastasis treatment. Recently, a new microfluidic chip for sorting and capturing single CTCs and CTMs has been demonstrated by Kulasinghe et al. [106], which is a promising new development in CTC/CTM utility. Another current approach involves using CTCs as a treatment target. Tumor metastasis is initiated by a CTC subpopulation which escaped from immune surveillance [107]. The expression of a number of immune checkpoints contribute essentially to the immune-editing capability of the CTCs. Thus, simultaneous blocking of those CTC checkpoints can be an immunotherapeutic approach. In fact, Lian et al. have succeeded in targeting both PD-L1 and CD47 on CTC surface, resulting in significantly lessened tumor growth and metastasis [108]. 

So far, there is a difference between the results of in vitro drug screening on cancer cell lines and the in vivo clinical efficacy. The lack of high throughput models with similarities to human cancer pathology leads to the low success rate of drug discovery in oncotherapy [109]. Recently, an emerging direction is using organoids to create novel cancer models, which would better resemble the real tumor both genotypically and phenotypically. Organoids are derivatives of 3D cancer cell culture including architectural and physiological similarities with the native tumor. The in vitro and in vivo cancer models using organoids can bridge the traditional preclinical models and clinical trials with the advantages of being less time-consuming, more economic, and more effective [110]. Seidlitz et al. [111] generated human and mouse GC organoids which modeled typical characteristics and altered pathways of human GC. These models revealed valuable information on the native tumor which can be used for drug screening and biomarker identification, thereby enabling precision medicines. In this direction, the idea of using CTC-derived organoids instead of tumor-derived organoids is tempting because of the advantages of real-time monitoring and a non-invasive nature. Bartucci et al. [112] described an interdisciplinary protocol to develop patient-derived organoids by using chimeric antigen receptor T cell (CAR-T) immunotherapy. Such an approach of patient specific therapy that targets cancer vulnerabilities, when combined with anti-proliferative and immunotherapeutic regimens, could fortify the power of cancer precision medicine. Another perspective in personalized medicine with immunotherapy is the application of microsatellite instability (MSI) analysis of CTCs. High MSI tumors which are associated with germline mutation in mismatch repair genes account for about 22% of GCs [113]. Advanced GC patients with high MSI had positive responses to pembrolizumab (ORR 85.7%) [114]. Although MSI analysis on GC CTCs has not been reported, Steinert et al. found mutational profiles of CTCs in colorectal cancer patients similar but not identical to the primary tumor tissue. Despite the mutation of some key genes such as KRAS or TP53 was not identified in the tumor, they were detectable in CTCs [115]. Such findings suggest a study on association of the high MSI CTCs with checkpoint immunotherapy response in GC.

One remaining question is whether there is a difference between molecular characteristics of CTCs derived from primary gastric tumor and those of various organ metastases-derived CTCs. Such information would be very useful for early detection of micro-metastasis and making rational treatment decisions in metastasis GC. In a similar way, a unique “CTC gene signature”, which is distinct from primary breast tumor and involves in signaling pathways associated with breast cancer brain metastasis (BCBM), was found by Boral et al., suggesting new application of CTCs in management of BCBM [116].

## 5. Conclusions

Despite the extensive efforts in improving GC treatment, therapy resistance remains a great challenge. One of the potential solutions to this problem is liquid biopsy, a less invasive tool which can provide real-time insights into the tumor. Various blood-based surrogate markers including circulating DNA, miRNA, extracellular vesicles, platelets, and CTCs have been studied for liquid biopsy in order to predict responders to a certain therapy [50,117]. Among these, CTCs, with their newly discovered clinical significance, could contribute additionally to the perspective of cancer precision medicines from prognosis to real-time monitoring, therapeutic targets, and drug discovery (Figure 2).

## Figures and Tables

**Figure 1 cancers-12-00695-f001:**
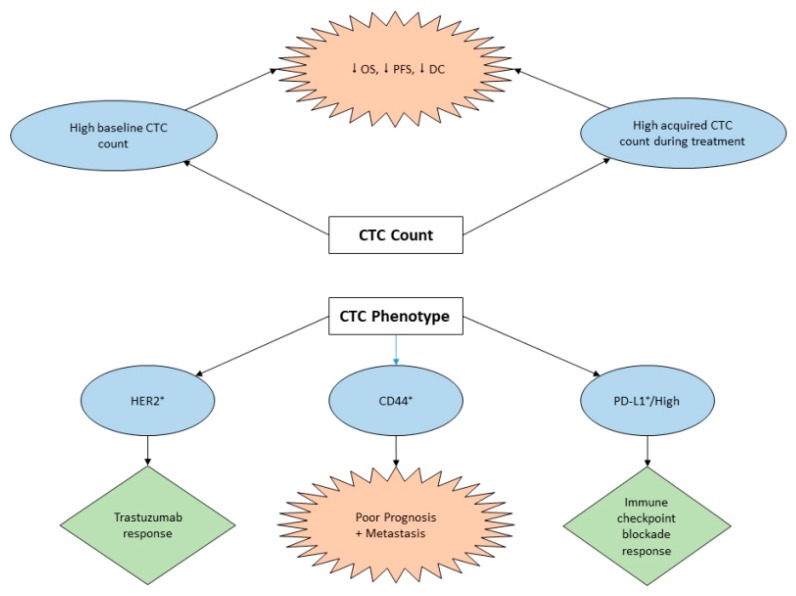
The predictive utility of CTCs in gastric cancer treatment. High baseline or acquired CTCs count was associated with decreased OS, PFS, and DC. The HER2 and PD-L1 positivity or high amplification could predict positive response to trastuzumab and checkpoint therapy, respectively. Meanwhile, the CD44^+^ CTCs were associated with poor prognosis and metastasis.

**Figure 2 cancers-12-00695-f002:**
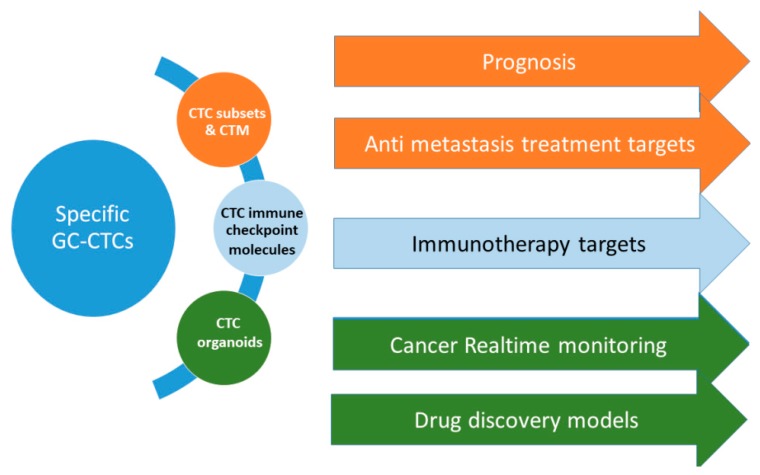
The future perspectives of CTCs in gastric cancer. Understanding GC-CTCs signatures would facilitate the use CTC subpopulations and CTMs in GC prognosis and as potential targets in anti-metastasis treatment; the use of immune checkpoint molecules on CTC surface as targets for GC immunotherapy; and the use of CTC organoids in cancer real-time monitoring and as in vitro and in vivo models for drug discovery.

**Table 1 cancers-12-00695-t001:** Systematic Review of the Clinical Significances of Circulating Tumor Cells (CTCs) in Gastric Cancer (GC) in Studies during 2015–2019 Period.

Author, Country, Year	Cases	Isolation and Identification Tool	Markers	OS and PFS of CTC(+) vs. CTC(-)	Positive Cutoff	Clinical Implications	Reference
Lee (South Korea), 2015	100 metastatic GC patients	anti-EpCAM antibody coated magnetic particles CTC-Profiler (Veridex)	EpCAM, CK8/18/19, CD45	OS: 120 days vs. 220 days; *p* = 0.03) PFS: 59 days vs. 141 days; *p* = 0.004	≥ 5CTCs/7.5 mL	CTCs are associated with poor response to chemotherapy in metastatic gastric cancer patients. CTC positivity was an independent adverse factor for PFS and OS.	[55]
Okabe (Japan), 2016	136 advanced GC patients	semi-automated immunomagnetic separation system CellSearch	EpCAM, CK8/18/19CD45, DAPI	OS: HR 2.20 [95%CI: 1.120–4.03]; *p* = 0.009 PFS: HR 2.03 [95%CI: 1.13–3.66]; *p* = 0.016	≥ 1 CTCs/7.5 mL	Detection of CTCs was an independent predictor of a shorter PFS in advanced gastric cancer. Patients who require intensive treatment: CTCs could be a valuable biomarker. The combined status of CTC and CY would be useful in selecting patients for radical surgery.	[56]
Zhou 2016	1110 GC patients in meta-analysis	-	-	OS: HR = 2.23, 95% CI: 1.86–2.66 PFS: HR = 2.02, 95% CI: 1.36–2.99	-	High CTCs count was associated with depth of infiltration regional lymph nodes metastasis and distant metastasis. For un-resectable GC patients, high CTCs count before and during chemotherapy was significantly correlated with poor OS, PFS, and DC rate.	[57]
Mishima(Japan), 2017	101 GC patients 15 advanced GC patients whose primary tumors were HER2-, but CTCs were HER2+	both 3 D-IF-FISH method & CellSearch System3D-IF-FISH only	EpCAM, CK8/18/19, CD45 HER2	OS and PFS of 15 advanced GC patients with CTC- HER2+: 6.1 months (95% CI: 2.1–10.0) and 14.4 months (11.0–17.8), respectively	≥ 1 CTCs/7.5 mL	New, non-invasive strategy to select patients who are likely to benefit from trastuzumab-based therapies, despite their primary biopsy being HER2-negative.	[58]
Liu (China), 2017	59 GC patients of stage II-IV	CELLection™ Epithelial Enrich kit	EpCAM, CK8/18/19, DAPI	OS: HR = 3.59, 95% CI:1.655-7.817, *p* = 0.001 PFS: = 2.81, 95% CI:1.313-5.999, *p* = 0.008	≥ 2 CTCs/5 mL	The baseline CTC count of >2 cells/5 mL and an increase of the CTC count after the first cycle of chemotherapy was an independent prognostic marker of poor PFS and OS→ patients with a low baseline CTC count or decrease of the CTC count after the first cycle of chemotherapy may benefit significantly from palliative chemotherapy	[59]
Zheng (China), 2017	81 GC patients	ISET-immunofluorescence	CK8/18/19, vimentin	CTM positivity was an independent factor for determining the PFS (*p* = 0.016) and OS (*p* = 0.003) of stage IV patients CTM correlated with shorter PFS and OS than single CTCs (*p* < 0.05)	≥ 1 CTCs/5 mL For CTM: ≥ 3 CTCs	In stage IV patients, CTM positivity was correlated with serum CA125 level. CTM were an independent predictor of shorter PFS and OS in stage IV patients. → CTM detection may be a useful tool to predict prognosis in stage IV patients.	[60]
Kang (South Korea), 2017	116 patients with gastric cancer patients & 31 healthy volunteers	“FAST disc” centrifugal microfluidic system	EpCAM, CK8/18/19, CD45 DAPI	-	≥ 2 CTCs/7.5 mL Sensitivity: 85.3 Specificity: 90.3	Although the clinical feasibility of CTCs for gastric cancer staging was not proved, these results suggest a potential role of CTCs as an early diagnostic biomarker of gastric cancer.	[61]
Yue (China), 2018	35 patients with different advanced gastrointestinal tumors	Pep MNPs isolated system	CK19, CD45 DAPI, PD-L1	PFS based on baseline PD-L1^high^ CTC count: 4.27 vs. 2.07 months HR = 3.342; 95%CI 1.488–7.505; *p*= 0.002 PFS based on post-therapeutic PD-L1^high^ CTC count: 3.4 vs. 2.1 months; HR= 0.412; 95%CI 0.177–0.962;, *p*= 0.031)	≥ 2 PD-L1^high^ CTCs/4 mL	The abundance of PD-L1^high^ CTCs at baseline might serve as a predictor to screen patients for PD-1/PD-L1 blockade therapies. Measuring the dynamic changes of CTC could indicate the therapeutic response at early time.	[62]
Yang (China), 2018	40 GC patients	wedge-shaped microfluidic chip (CTC-ΔChip) & three-color immunocytochemistry method	(CK, CD45, Nucleus marker	-	-	CTC-ΔChip exhibited the feasibility of detecting CTCs from different types of solid tumor, and it identified 7.30 ± 7.29 CTCs from 2 mL peripheral blood with a positive rate of 75% (30/40) in GC patients. Novel CTC-ΔChip shows high performance for detecting CTCs from less volume of blood samples of cancer patients and important clinical significance in GC.	[63]
Li (China), 2018	115 advanced GC patients, including 56 tumor HER2^+^ subjects who received first-line HER2-targeted therapy plus chemotherapy and 59 tumor HER2^−^ subjects who received chemotherapy alone	IF-FISH Cytelligen system	DAPI, HER2, CEP8, and CD45	-	-	CTC HER2^+^ was found in 91.0% of tumor HER2^+^ and 76.2% tumor HER2^−^ patients and was correlated with development of resistance to trastuzumab for the tumor HER2^+^ patients and chemotherapy alone for the tumor HER2^−^ patients. Determining of CTC HER2 showed advantages in real-time monitoring of therapeutic resistance.	[64]
Cheng (China), 2019	32 advanced GC patients	CanPatrol CTC enrichment technique Multiplex RNA in situ hybridization assay	EpCAM, CK8/18/19, CD45 DAPI, PD-L1, Vimentin and Twist	-	≥ 2 PD-L1^+^ CTCs/5 mL	CTCs count was well correlated with clinicopathology parameters. Enumeration of epithelial CTC subset and its relative abundance in the total CTC pool are highly correlated with clinical efficacy. Monitoring CTC subtypes exhibits higher sensitivity of evaluating the disease status, compared to the traditional methods.	[65]
Lu (China), 2019	42 GC patients of stage III-IV	ISET-ICC method followed by IHC	EpCAM, CK8/18/19, CD45, Vimentin and Twist, E-cadherin	-	-	The threshold number of CTCs is significantly associated with different clinical stages and was positively correlated with the value in U/mL of CA724. CTCs technology based on ISET method has a high detection rate. CTCs are promising predictor for the evaluation and prediction of treatment responses in stage III–IV gastric cancer.	[66]
Abdallah (Brazil), 2019	At diagnosis (55 samples before neoadjuvant treatment) After surgery and before adjuvant therapy (33 samples)	ISET and immunocytochemistry & microscopy	HER2 and plakoglobin, CD45	-PFS between CTM-positive patients vs. CTM-negative patients (18.7 months vs. 21.6 months; *p* = 0.258 -PFS between plakoglobin-positive CTM patients vs. plakoglobin-positive CTM patients: 15.9 months vs. 21.3 months; *p*= 0.114	≥ 1 CTM (2 CTCs)/4 mL	The analysis of CTM plakoglobin expression is a promising tool in the understanding the biology and prognosis of GC.	[67]
Gao, 2019	3814 GC patients in meta-analysis	-	-	HR = 1.84, 95%CI 1.50–2.26, *p* < 0.001	-	CTC positivity was associated with poorer OS.	[68]

CellSearch: semi-automated immunomagnetic separation system; CTM: circulating tumor microembolies; FISH: fluorescent in situ hybridization; IF: immunofluorescence; ISET: isolation by size of epithelial tumor cells; ICC: immunocytochemistry; IHC: immunohistochemistry; OS: overall survival; PFS: progression-free survival.

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
