# Peer review of "Emerging Role of Circulating Tumor Cells in Gastric Cancer"

_cancers, 2020, doi:10.3390/cancers12030695_

Round 1

Reviewer 1 Report

The authors reviewed the studies about CTCs of GC and showed that they have potential as novel biomarkers and/or non-invasive method for diagnostics, prognostics and treatment monitoring of GC.

  1. The author mentioned that the induction of EMT is the most important to survive CTCs in the blood vessels and metastatic site. How does mesenchymal phenotype help CTCs to survive in such microenvironments?

  1. What is the significance of “self-seeding”?

  1. Is it possible that CD44(+)/CD45(-) gastric CTCs are cancer stem cells?

  1. What are the targets of miRNAs (miR-421, miRNA543, miR106a, and miR17) detected in the GC patients?

  1. Is it possible to examine microsatellite instability of CTCs? MSI-h GCs are good candidates of anti-PD-1 therapy.

  1. What is the difference between CTCs and cell free DNA analyses?

Author Response

Reviewer #1

Question 1

The author mentioned that the induction of EMT is the most important to survive CTCs in the blood vessels and metastatic site. How does mesenchymal phenotype help CTCs to survive in such microenvironments.

Answer 1

We thank the reviewer for pointing out this and have added more information regarding this important feature in Page 5, lines 140-144.

Question 2

What is the significance of “self-seeding”?

Answer 2

We have added the significance of “self-seeding” as follow (Page 5, lines 147-153).

 “…Self-seeding contributes towards metastatic development, because due to the leaky neovasculature of tumors, “self-seeding” CTCs in circulation would not require significant adaptations to infiltrate their host tumor [32]. Consequently, the host tumor would we enriched with aggressive CTCs that would lead to an increase in the overall metastatic potential of the tumor. It was found by Kim et al., in a study of MDA231 mammary tumors in mice, that self-seeded tumors had a significant increase in the rate of tumor growth, along with increased angiogenesis and stromal recruitment [32]…”

 Question 3

Is it possible that CD44(+)/CD45(-) gastric CTCs are cancer stem cells?

Answer 3

While there have been as many as evidence the role of CD44+ as a cancer stem cells marker, few, however, provided information of CD45(-).

We added citation #74 (Page 9, lines: 308-309) reported that, CD44(+)/CD45(-) was only mentioned as marker for GC CTCs with high malignancy.

Question 4

What are the targets of miRNAs (miR-421, miRNA543, miR106a, and miR17) detected in the GC patients.

Answer 4

We thank the reviewer for pointing out this. We have added targets of the miRNAs, as suggested by the reviewer, in Page 7, lines 263-270.

Question 5

Is it possible to examine microsatellite instability of CTCs? MSI-h GCs are good candidates of anti-PD-1 therapy

Answer 5

We thank the reviewer for this suggestion. We have added this information on the perspectives of using MSI-h CTC in GC checkpoint therapy in Page 11, lines 419-427.

Question 6

What is the difference between CTCs and cell free DNA analyses

Answer 6

We have added comments on advantages and disadvantages of cfDNA vs CTCs and the perspectives of combining the 2 markers in Page 11, lines 383-390.

Reviewer 2 Report

This is a concise review describing the clinical utility of circulating tumor cells in gastric cancer management.  It is clearly written.  It would have been nice if the authors would have commented more extensively on additional "liquid biopsy" components, such as the emerging data on miRNA, platelets, proteomics, exosomes, etc. These blood components, in addition to the prognostic value of CTCs, will really be the future of precision medicine.

Author Response

Reviewer #2

Question 1

It would have been nice if the authors would have commented more extensively on additional "liquid biopsy" components, such as the emerging data on miRNA, platelets, proteomics, exosomes, etc. These blood components, in addition to the prognostic value of CTCs, will really be the future of precision medicine.

Answer 1

We thank the reviewer for this suggestion and have added comments on the context of liquid biopsy with other blood-based markers in Page 11, lines 435-440.

Round 2

Reviewer 1 Report

  1. It is uncertain that PD-L1 expression in CTC become a biomarker to predict the efficacy of immunecheck point inhibitor because only one article was published (ref.91). 
  2. According to your systemic review on CTC in gastric cancers, it is too early to conclude that analysis of CTC is useful for precision medicine.

Author Response

Question 1

It is uncertain that PD-L1 expression in CTC become a biomarker to predict the efficacy of immunecheck point inhibitor because only one article was published (ref. 91).

Answer 1

We agree with reviewer comment. We first revise the text describing that CTC or CTC PD-1/PD-L1 expression may be potential predictive biomarkers for cancer immunotherapy efficacy. We cite more articles discussing about this suggestion.  We then propose that applying –omics and next-generation sequencing (NSG) may provide a solution to validate CTC PD-1/PD-L1 as a predictor of immune checkpoint blockade therapy. Please see these revised parts on Page 10, line 356-361 and line 369-376.

 Question 2

According to your systemic review on CTC in gastric cancer, it is too early to conclude that analysis of CTC is useful for precision medicine.

Answer 2

We thank the reviewer for this suggestion and have revised the last sentence in conclusion as follow (Page 18, lines 453-455):

“…Among which, CTCs, with their newly discovered clinical significance would contribute additionally to the perspective of cancer precision medicines, from prognosis, real-time monitoring to therapeutic targets and in drug discovery as well (Figure 2)…”
